# Reanalysis of the 2000 Rift Valley fever outbreak in Southwestern Arabia

**Compton J. Tucker**[1]*, **Katherine A. Melocik**[1], **Assaf Anyamba**[1], **Kenneth J. Linthicum**[2], **Shamsudeen F. Fagbo**[3,4], **Jennifer L. Small**[1]

**1** Earth Sciences Division, NASA/Goddard Space Flight Center, Greenbelt, Maryland, United States of America, **2** Center for Medical, Agricultural, and Veterinary Entomology, U.S. Department of Agriculture, Gainesville, Florida, United States of America, **3** One Health Unit, Executive Directorate for Response and Surveillance, National Centre for Disease Prevention and Control, Riyadh, Saudi Arabia, **4** Department of Public Health, Nigerian Institute of Medical Research, Yaba, Lagos, Nigeria

* compton.j.tucker@nasa.gov

**Data Availability Statement:** Data are available at Zenodo: (https://doi.org/10.5281/zenodo.4133832).

## Abstract

The first documented Rift Valley hemorrhagic fever outbreak in the Arabian Peninsula occurred in northwestern Yemen and southwestern Saudi Arabia from August 2000 to September 2001. This Rift Valley fever outbreak is unique because the virus was introduced into Arabia during or after the 1997–1998 East African outbreak and before August 2000, either by wind-blown infected mosquitos or by infected animals, both from East Africa. A wet period from August 2000 into 2001 resulted in a large number of amplification vector mosquitoes, these mosquitos fed on infected animals, and the outbreak occurred. More than 1,500 people were diagnosed with the disease, at least 215 died, and widespread losses of domestic animals were reported. Using a combination of satellite data products, including 2 x 2 m digital elevation images derived from commercial satellite data, we show rainfall and potential areas of inundation or water impoundment were favorable for the 2000 outbreak. However, favorable conditions for subsequent outbreaks were present in 2007 and 2013, and very favorable conditions were also present in 2016–2018. The lack of subsequent Rift Valley fever outbreaks in this area suggests that Rift Valley fever has not been established in mosquito species in Southwest Arabia, or that strict animal import inspection and quarantine procedures, medical and veterinary surveillance, and mosquito control efforts put in place in Saudi Arabia following the 2000 outbreak have been successful. Any area with Rift Valley fever amplification vector mosquitos present is a potential outbreak area unless strict animal import inspection and quarantine proceedures are in place.

## Introduction

Rift Valley fever is a mosquito-borne viral hemorrhagic fever that affects domestic animals and humans. First described in 1931, the virus responsible for Rift Valley fever is in the genus *Phlebovirus* and the family *Bunyaviridae*. The first documented occurrence or epizootic of Rift Valley fever occurred in 1930–1931 near Lake Naivasha, Kenya, in the Rift Valley of East Africa, with high mortality among domestic sheep [1]. Although Rift Valley fever primarily affects domestic animals, mainly sheep, goats and cattle, it can also infect humans, causing illness and

**Funding:** AA: U.S. Army Medical Command contract DHA-2018-770 for Global Emerging Infectious Disease Surveillance.

**Competing interests:** The authors have declared that no competing interests exist.

death in some cases [2]. Most human infections have resulted from bites from infected mosquitoes, through contact with infected animals, and to a lesser extent, through contact with infected animal tissue [3]. Rift Valley fever has been exclusively a disease of Africa and Madagascar, with the exception of the Southwest Arabian 2000–2001 outbreak [4, 5] (Fig 1).

Rift Valley fever infections in humans are usually mild, with an incubation period of 2 to 5 days following exposure. Those infected may experience no detectable symptoms or may develop a sudden-onset influenza-like fever, headache, gastrointestinal discomfort, and general body malaise. However, these symptoms are transitory and do not persist for more than 7 days. Other patients develop symptoms that may be mistaken for meningitis: loss of appetite; vomiting; sensitivity to light; and neck stiffness. While most human Rift Valley fever infections are mild, ~4% of human infections are more severe. These more severe infections are manifested as one or more of three syndromes: ocular disease; meningoencephalitis; and/or hemorrhagic fever [2].

While death is uncommon for patients with the ocular form of Rift Valley fever, some infected patients experience a permanent loss of vision. The meningoencephalitis form of Rift Valley fever is usually manifested a few weeks after initial infection, with symptoms including convulsions, lethargy, coma, hallucinations, memory loss, and vertigo. The death rate of this form of the disease is low, although neurological complications can appear several weeks later. The hemorrhagic fever form of Rift Valley fever has a case fatality rate of ~50%. Symptoms of this form of the disease include bleeding in the gastrointestinal tract, bleeding in the skin, bleeding from the nose or gums, other hemorrhage, and severe liver impairment. Death is reported to occur 3 to 6 days after onset of symptoms of this severe form of Rift Valley fever. Fortunately, this Rift Valley fever syndrome is infrequent in infected humans [2].

There is no licensed Rift Valley fever vaccine for humans. Several effective vaccines for animals are available; however, animal immunization must occur before an outbreak starts. Thus, early warning of possible pending outbreaks several weeks in advance is extremely important, as this enables vaccination programs for animals, enables personal protection methods for humans, and enables mosquito vector control to be undertaken.

## Mosquito vector ecology and population dynamics

Rift Valley fever virus transovarially infects the eggs of at least one species of the mosquito genus *Aedes* [6]. That is, the mosquito eggs are laid already infected with the virus. These infected eggs lie dormant in the soil of unflooded mosquito habitats in the endemic area. Following periods of prolonged rainfall, local flooding occurs and the already-infected eggs hatch and immature mosquito stages develop into infected adults. These infected mosquitoes transmit the Rift Valley fever virus to animals upon whom they feed, although usually at a very low level, because they are inefficient transmitters of the disease [7]. Among domestic animals, there is variability in susceptibility of different species and breeds with sheep, goats, and camels being the primary species affected. Although larger wild animals become infected and develop viremia, most wild animals in endemic regions exhibit minimal disease [8, 9].

Without prolonged rainfall for several weeks, any initial Rift Valley fever outbreak would be local in nature and would not be a reported outbreak. However, if excessive rainfall continues for 5 to 7 weeks duration or longer, mosquitoes of the genus *Culex* develop in these flooded areas, hatch, become numerous, and are very efficient transmitters of the Rift Valley fever virus (Fig 2). The *Culex* and *Mansonia* species are the amplification or expansion vectors for Rift Valley fever outbreaks. They acquire the virus from infected livestock and then efficiently infect large numbers of other animals quickly [4, 10, 11].

Rift Valley fever has occurred intermittently over much of sub-Saharan Africa, with outbreaks reported in many countries (Fig 1). In East and Southern Africa, Rift Valley fever

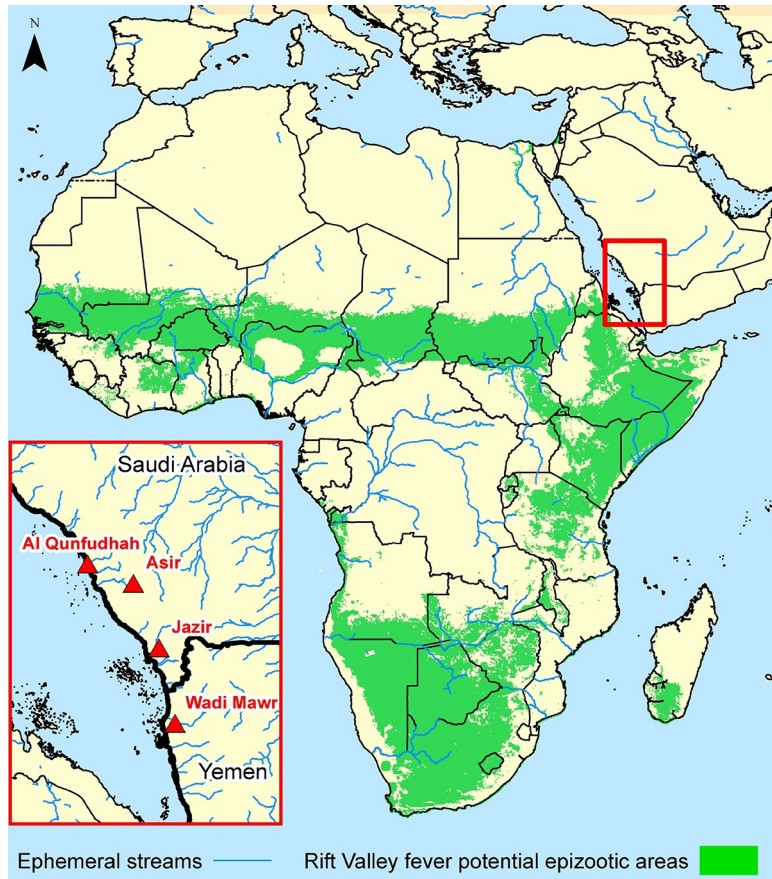

**Fig 1. Potential epizootic areas of Rift Valley fever in Africa and Madagascar are shown in green [4].** Four areas in Saudi Arabia and Yemen are shown in the insert where significant numbers of people were infected in the 2000–2001 outbreak were noted by the red triangles. While Rift Valley fever has largely been an African disease, there were no documented Rift Valley fever outbreaks in Arabia prior to 2000–2001 and there have been none since that time.

activity has been reported in the proximity of grassland depressions [12–15]. These depressions can flood seasonally and, when flooded, are good habitats for the development of immature *Aedes* and *Culex* mosquito species that serve as Rift Valley fever vectors [10, 11]. In dry areas of Senegal and Mauritania and in irrigated areas of Egypt, Rift Valley fever outbreaks have occured in riverine areas [16–18].

## Area of study

Our area of study was the southwestern part of the Arabian Peninsula, where Saudi Arabia borders Yemen along the Red Sea (Fig 3). There are three climatic zones present in this region: the Tihama, the dry coastal plain with elevations below 400 meters above sea level and minimal precipitation, usually <150 mm/yr; a region composed of hills with elevations between 400 and 600 meters with <300 mm annual precipitation; and a mountainous region with elevations 600 meters and above with annual precipitation of 300 to 800 mm annual precipitation [19]. The Tihama is a large alluvial plain that is intensely cultivated where irrigation is possible from many dams. Periods of higher precipitation in the Sarawat Mountains result in higher runoff into ephemeral streams that flow to the Tihama. These agricultural areas and reservoirs create pools of standing water ideal for mosquito production, particularly during periods of excessive and persistent rainfall upstream.

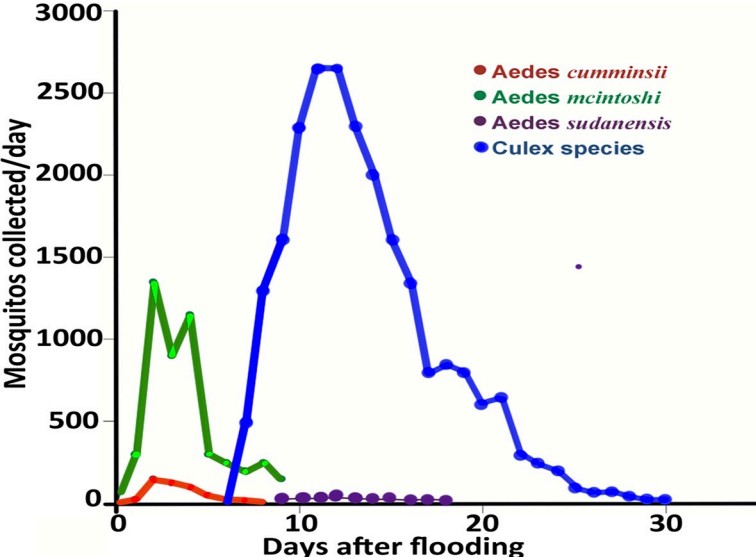

**Fig 2. Mosquito numbers after mosquito habitats were flooded in East Africa as determined from collecting and identifying mosquito species.** Vector specimens collected from the Kamiti dambo, Kenya from November 28— December 27, 1982 [6].

Remote sensing surveillance of Rift Valley fever is based upon: (1) Monitoring through time rainfall or a surrogate for rainfall in the Rift Valley fever endemic area (Fig 1); and (2) Identifying periods of anomalous rainfall that persist for more than 5 to 7 weeks in the Rift Valley fever endemic area. Previous work using satellite and ground data has documented and confirmed this approach for several areas [11, 12, 14, 20].

We use the satellite normalized difference vegetation index (NDVI) [21] as a surrogate for rainfall. We also introduce a new capability for mapping mosquito breeding habitats, through the use of digital elevation data derived from commercial satellite data stereo image pairs [22]. This new mapping of possible mosquito breeding areas at the 2 x 2 m scale identifies potential inundation areas.

There are two methods commonly used to monitor or infer rainfall from Rift Valley fever endemic areas: (1) Measure rainfall directly using the rain gauge network and/or use rainfall estimates from satellite data such as the Tropical Rainfall Monitoring Mission [23], geostation-ary MeteoSat satellites [24], or the Global Precipitation Mission [25] or (2) Use a NDVI remote sensing approach from which rainfall can be inferred [26–28]. While it would be desirable to monitor rainfall directly, the rain gauge network in much of sub-Saharan Africa is sparse and discontinuous [28] and estimates from Meteosat, the Tropical Rainfall Monitoring Mission, and the Global Precipitation Mission have only been available since the early 1990s and have a coarse spatial resolution. For these reasons, and especially for spatial specificity at the scale of tens to hundreds of meters, we use the time series NDVI-rainfall monitoring approach.

NDVI time series exist from the Advanced Very High Resolution Radiometer (AVHRR) instruments since July 1981 globally at an 8-km resolution [29]; from the Satellite Pour l'Observation de la Terre (SPOT) vegetation-PROBA V-Sentinel 3 satellite continuum and have been available globally since May 1998. SPOT vegetation data have a spatial resolution of 1 km from 1998–2013 [30] while PROBA V and Sentinel-3 data are available at 330 m from 2013 [31]; from NASA's moderate resolution imaging spectrometer (MODIS) instruments availably globally since January 2000 at a 250 m resolution [32]; and from Landsat at 30 m since Landsat-4 was launched in 1982 [33].

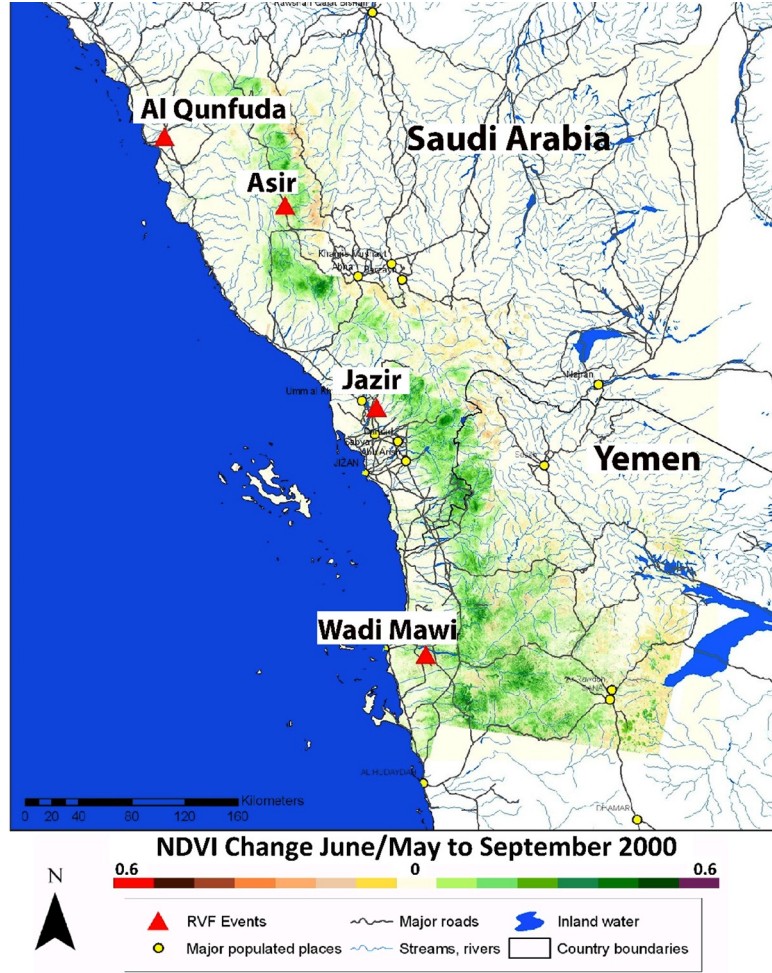

**Fig 3. The Landsat-7 NDVI difference between May/June 2000 and September 2000 shows the increase in green vegetation density from increased rainfall by September 2000 in the Sarawat Mountains for Southwest Arabia.** This in turn, resulted in more runoff to the coastal plain or Tihama that provided ideal conditions for the emergence of Rift Valley fever mosquito vectors. Specific outbreak locations occurred from Al Qunfuda, Saudi Arabia in the north to Wadi Mawr, Yemen in the south and are noted by red triangles where they were reported. This figure is from Anyamba et al. [5].

The NDVI approach we employ is based upon spectral characteristics of vegetation directly related to photosynthesis in terrestrial vegetation. Green leaves in plant canopies have low red reflectance due to strong chlorophyll absorption in the red portion of the spectrum and a very high reflectance in the near infrared portion, where minimal absorption occurs. This unique spectral response of green leaves makes it possible to differentiate green vegetation from other surface constituents by remote sensing. The normalized difference vegetation index is calculated as:

$$NDVI = \frac{\rho_{nir} - \rho_{red}}{\rho_{nir} + \rho_{red}} \qquad (1)$$

Where $\rho_{nir}$ and $\rho_{red}$ are the surface reflectances in the near infrared and red portions of the electromagnetic spectrum, respectively [19]. Derived NDVI values range between -1 to +1, with values below zero indicating absence of green vegetation and positive values progressively

showing more green leaf density. The variability of rainfall and the dynamics of vegetation in semi-arid lands are a major determinant of life cycles of animals and insects. There is a close relationship between seasonal green vegetation density and persistence, the NDVI, and breeding and upsurge patterns of particular insects, including locusts and mosquitoes [12, 13, 15, 16, 34, 35].

In addition to using NDVI data from a variety of different instruments to infer rainfall from subsequent green vegetation development, we employ digital elevation images derived from commercial satellite data along-track stereo image pairs to map the potential for inundation at the 2 x 2 m scale in Rift Valley fever outbreak areas [22]. The area of possible inundation is directly related to potential mosquito breeding density.

## The Southwestern Arabian 2000 Rift Valley fever outbreak

Reports of an outbreak of an unknown disease first appeared in Saudi Arabian newspapers in early September 2000. Initial reports in mid-September 2000 were from Jazan state, in Southwestern Saudi Arabia adjacent to Yemen, and described an unknown disease infecting ~ 60 people with ~30 fatalities [36]. Disease symptoms in the affected patients were described as high fever, diarrhea, burst arteries, and renal failure. Bleeding from the mouth, nose, and ears and eye lesions were common in affected patients. Jazan state is known for its heat and humidity during the rainy season at this time of year, that makes it an excellent breeding area for mosquitoes. A spokesman for the Saudi Arabian health authority reported the disease was transmitted by sheep and a large number of domestic animals had died in Jizan state, Saudi Arabia [36]. Jazan state lies 300 km across the Red Sea from Africa and is adjacent to Yemen.

Saudi Arabian authorities in Jazan State imposed emergency measures after cases of the Rift Valley fever were confirmed in late 2000. Measures included closing schools for several weeks, intensive mosquito control spraying, and a ban on the importation of domestic animals from the Horn of Africa.

On September 20, 2000, another Rift Valley fever outbreak was reported in the Wadi Mawr area of the Huseidah region of Yemen that resulted in 50 deaths [37]. Laboratory confirmation of the disease in both Saudi Arabia and Yemen was made by the World Health Organization's Collaborating Centre at the Centers for Disease Control and Prevention in Atlanta, Georgia [38].

By early November 2000, more than 1,500 people were diagnosed as having contracted Rift Valley fever in both countries and at least 215 died: 106 in Saudi Arabia and 109 in Yemen. The death rate is perhaps the highest case fatality rate (~13%) for a Rift Valley fever outbreak to date. However, the number of deaths could be correct while the number of infected people was under reported, thus lowering the case fatality rate. Several thousand domestic animals also perished but exact numbers were not reported [39].

## Materials and methods

Our analysis was preceded by conference presentation in 2014 that used ecological niche modeling coupled with satellite-derived land surface temperature and NDVI [40]. This work was calibrated based on previous African outbreaks and extrapolated to predict outbreak sites in Arabia in the Gizan region. We provide additional evidence for the validity of this approach by using three sources of NDVI data that identified the same outbreak areas in western Saudi Arabia. Three sources of NDVI satellite data were used for our rainfall analysis in southwestern Arabia: 8-km data from 1981 to 2016 derived from the AVHRR instruments carried on a series of polar-orbiting meteorological satellites operated by the National Oceanic and Atmospheric Administration and the European Organization for the Explotation of Meteorological

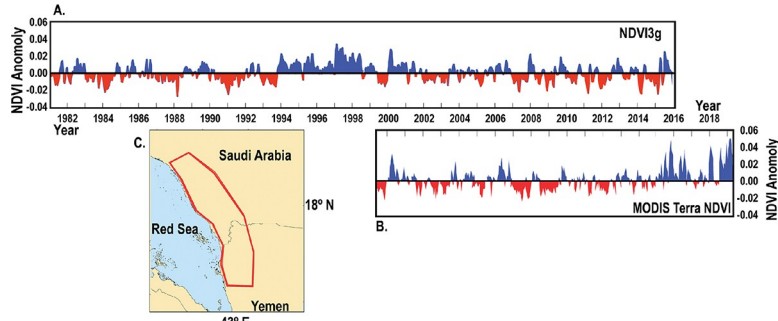

**Fig 4.** (A). The AVHRR NDVI 8 km anomalies from July 1981 to December 2016 from the area in (4c). (B). The MODIS 250 m NDVI anomalies for the same area from 2000 to 2019 in (4c) from Al Qunfunda, Saudi Arabia to Wadi Mawr, Yemen, from the coast to the foothills of the Sarawat mountains, an area of 100,000 km² in Southwestern Arabia bordering the Red Sea in the image of (C). Positive NDVI values are a consequence of above-average rainfall and are color-coded in blue in both 4a and 4b [27]. Dryer periods are color-coded in red. It is possible the introduction of Rift Valley fever virus into Southwest Arabia occurred in 1997–1998, while the East Africa 1997–1998 Rift Valley fever outbreak was underway. Note the very similar NDVI trends from 2000 to 2016 derived from AVHRR NDVI and from MODIS NDVI data. The MODIS NDVI data in Fig 4b show favorable conditions for Rift Valley fever outbreaks in 2016–2019 but none have occurred in Arabia during this time.

Satellites; NASA 250 m MODIS imagery from 2000 to early 2020; and NASA and US Geological Service 30 m data derived from Landsat-7's Enhanced Thematic Mapper instrument for 2000 and 2001. We now are able to complement Landsat 30 m data with "harmonized" Landsat-8 and Sentinel-2a and Sentinel-2b 30 m data if greater spatial detail is needed [41]. These harmonized data provide time series NDVI data at a 30 m spatial resolution with a revisit frequency of 3.7 days at the equator. Fig 4 shows the NDVI correspondence between AVHRR 8 km and MODIS 250 m data for the Arabian outbreak area.

We also used two DigitalGlobe WorldView-1 along-track stereo pair images acquired on November 7, 2017 only seconds apart from which elevation data were determined. These data were processed using the Surface Extraction with TIN-based Search-space Minimization (SETSM) algorithm [22] to produce a 2 x 2 m digital elevation hillshaded image with a vertical uncertainty of ±70 cm (Fig 5). An analysis of the digital elevation profile and image for Fig 5 found 2.1 km² of potentially inundated areas that were comprised of 4,188 different 100 km² or greater non-connected areas. This capability is a new tool to address mosquito-linked infectious diseases and provides direct identification and innumeration of small mosquito breeding areas.

## Results and discussion

In tandem with the Landsat-7 NDVI analysis in Fig 3, we examined NDVI3g AVHRR data from 1981 to 2016 and MODIS 8-day NDVI from April 2000 through April 2019 (Fig 4). Because the outbreak occurred in September 2000, it must have rained in the region during the preceding weeks to allow for the flooding of mosquito breeding habitats and subsequent emergence of the Rift Valley fever vectors to initiate a large-scale outbreak [4]. The Landsat, NDVI3g, and MODIS NDVI anomalies from May to September 2000 showed a strip of persistent positive anomalies along the Sarawat mountain range. This mountain range, ~2000 meters above the sea and parallel to the Red Sea, is the most prominent elevated topographic feature in this region. Rainfall falling on the mountain range drains into ephemeral streams that in turn flow to the coastal plain. The reported focal areas of the outbreak in Saudi Arabia, Al Qunfuda and Jizan are cities, while Asir is a state. The reported focal area of the outbreak in Yemen was Wadi Mawr, also a district. These four locations are located on the coastal plain (Fig 3).

It has been suggested that 2000 Arabian Rift Valley fever outbreak resulted from the importation of infected livestock animals from East Africa before the 2000 outbreak. The period of rainfall from late August 2000 into 2001 would have provided the conditions for the presence of mosquito amplification vectors (Fig 4) [42–45]. However, it is not known if the virus was introduced to Arabia in 1997–1998 or 1999–2000.

However, the genetic sequence of the Rift Valley fever viruses isolated in Saudi Arabia during this period were closely related to virus samples isolated during the 1997–1998 epizootic in East Africa [46–48]. While no Rift Valley fever outbreaks were reported in Africa in the June to September 2000 time period, the disease can be present and be undetected in a residual manner within herbivore populations.

Most agriculture in this region is practiced in the coastal plain, and during the heavy rains in the Sarawat Mountains, agricultural fields become inundated, creating ideal habitats for mosquito breeding as we show with our 2 m scale digital elevation surface mapping. Malaria has been known to be endemic in the lowlands of this region and various species of *Anopheles* mosquitoes, including *Anopheles arabiensis* and *Anopheles gambiae*, have been identified there [49].

Other mosquito species possibly implicated in Rift Valley fever Southwest Arabian outbreak include *Culex tritaneniorhynchus* and *Aedes vexans arabiensis* mosquitoes in the Jizan area of Saudi Arabia, where more than 75,000 mosquitoes were collected between September 25, 2000 and October 10, 2000 and 23,699 were tested for Rift Valley fever virus [50].

Since early 2001, the Ministries of Agriculture and Health of Saudi Arabia have taken precautionary measures to prevent renewed outbreaks of Rift Valley fever. These measures included: providing Rift Valley fever animal vaccines and veterinarians to carry out the vaccination program; undertaking mosquito control efforts during periods of high rainfall; and routine mosquito testing for the Rift Valley fever virus [51]. Medical surveillance of humans has also been actively undertaken, with possible Rift Valley fever cases reported in September 2004 in Al Karbous village [52], in December 2008 in Asir state [53, 54], in May 2009 where 33 cases were reported in Jeddah and 103 in Mecca [55], and in May 2010 in Najran City [56]. Routine veterinary serosurveillance is also undertaken, with five seropositive Rift Valley fever cases detected in four flocks of sheep reported in April 2004 in Jizan, although without clinical confirmation [57].

Al-Afaleq et al. 2012 [58] report a summary of veterinary surveillance, where a total of 3,480 sheep, goats, cattle and camels with no history of vaccination against Rift Valley fever were tested at random in Saudi Arabia. All animals tested were negative for IgG class antibodies against the virus, except three out of 913 goats and four out of 1,508 sheep, which tested positive. No virus activity evidence was found in the areas studied where the three goats and four sheep tested positive. All seven animals were clinically normal. These rare positive cases were attributed to false positives or vaccinated animals transferred from other areas [58].

While Saudi Arabia has instituted a thorough, continuing, and apparently successful program of medical and veterinary Rift Valley fever surveillance since the 2000 outbreak, the same cannot be said for Yemen. The on-going conflict in Yemen has made animal importation inspections, quarantine, and medical and veterinary surveillance activities there impossible since 2015.

## Conclusions

The Arabian 2000 Rift Valley fever outbreak was unique in that infected animals were imported into areas where large numbers of mosquito amplification vectors were present, although it is not known when the infected animals were introduced into Arabia.

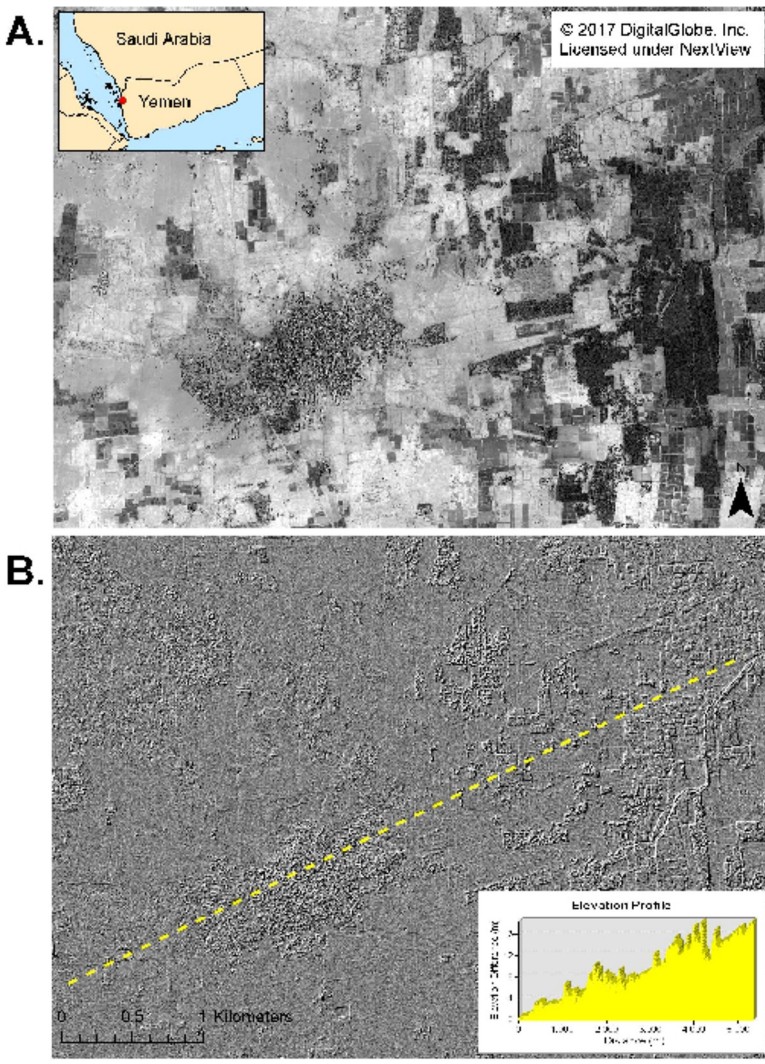

**Fig 5.** (A) WorldView-1 panchromatic 2 m resolution image from November 7, 2017 from the Wadi Mawr village area of Northwest Yemen; and (B) a stereo digital elevation hillshaded image and elevation profile produced from a stereo pair of WorldView-1 panchromatic images acquired on November 7, 2017 from the same area. The high-resolution elevation data shows potential inundated areas along the terrian. Mapping inundated areas to identify potential mosquito-breeding habitats is a new tool for infectious disease study.

Amplification vector mosquitos fed upon infected animals and the outbreak started. Using satellite data, we document the 2000 to 2018 precipitation regime for the coastal areas to the Sarawat Mountains of southwest Arabia with time series MODIS NDVI data at the 250 m scale in Fig 4. These data, coupled with the extensive number of potential-inundated mosquito breeding areas determined from digital elevation imagery at the 2 x 2 m scale in Fig 5, document excellent mosquito breeding potential in southwest Arabia in wetter periods. The fact no major RVF outbreak have occurred since 2000, and especially between 2015 to 2018 when the rainfall regime was very favorable to outbreaks, leads to two conclusions: (1) the Rift Valley fever reservoir was not established in Aedes mosquitos in Southwest Arabia after the 2000 outbreak; and/or (2) effective Rift Valley fever domestic animal quarantine and import restrictions, medical and veterinary surveillance, and mosquito control institutued by the Saudi Arabian Ministry of Agriculture have prevented outbreaks in that country. The conflict in

Yemen since 2015 hinders preventative action such as undertaken in Saudi Arabia and is thus a cause for concern.

## Acknowledgments

We wish to acknowledge the assistance of Mohammed Al-Hazmi from the Ministry of Health, Jizan, Saudi Arabia and Sami Mukhdari Mushta from the Saudi Center for Disease Prevention and Control, Riyadh, Saudi Arabia for assistance in both understanding the sequence of events in the 2000 Rift Valley fever outbreak and their description of the control and survellience methods within Saudi Arabia that have prevented subsequent outbreaks.

## Author Contributions

**Conceptualization:** Compton J. Tucker, Assaf Anyamba, Kenneth J. Linthicum.

**Data curation:** Jennifer L. Small.

**Formal analysis:** Compton J. Tucker, Katherine A. Melocik, Assaf Anyamba, Kenneth J. Linthicum, Jennifer L. Small.

**Funding acquisition:** Assaf Anyamba.

**Investigation:** Compton J. Tucker, Katherine A. Melocik, Assaf Anyamba, Kenneth J. Linthicum, Jennifer L. Small.

**Methodology:** Compton J. Tucker.

**Resources:** Kenneth J. Linthicum, Jennifer L. Small.

**Supervision:** Katherine A. Melocik.

**Validation:** Compton J. Tucker, Katherine A. Melocik, Shamsudeen F. Fagbo, Jennifer L. Small.

**Visualization:** Katherine A. Melocik, Shamsudeen F. Fagbo, Jennifer L. Small.

**Writing – original draft:** Compton J. Tucker.

**Writing – review & editing:** Compton J. Tucker, Katherine A. Melocik, Assaf Anyamba, Kenneth J. Linthicum, Shamsudeen F. Fagbo, Jennifer L. Small.

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
