## [Decision Letter · Decision Letter 0]

20 Sep 2019

PONE-D-19-21159

Reanalysis of the 2000 Rift Valley fever outbreak in Southwestern Arabia

PLOS ONE

Dear Dr. Tucker,

Thank you very much for submitting your manuscript "Reanalysis of the 2000 Rift Valley fever outbreak in Southwestern Arabia" (#PONE-D-19-21159) for review by PLOS ONE. As with all papers submitted to the journal, your manuscript was fully evaluated by academic editor (myself) and by independent peer reviewers. The reviewers appreciated the attention to an important health topic, but they raised substantial concerns about the paper that must be addressed before this manuscript can be accurately assessed for meeting the PLOS ONE criteria. Therefore, if you feel these issues can be adequately addressed, we invite you to submit a revised version of the manuscript that addresses the points raised during the review process. We can’t, of course, promise publication at that time.

We would appreciate receiving your revised manuscript by Nov 04 2019 11:59PM. To enhance the reproducibility of your results, we recommend that if applicable you deposit your laboratory protocols in protocols.io, where a protocol can be assigned its own identifier (DOI) such that it can be cited independently in the future. For instructions see: http://journals.plos.org/plosone/s/submission-guidelines#loc-laboratory-protocols

We look forward to receiving your revised manuscript.

Kind regards,

Abdallah M. Samy, PhD

Academic Editor

PLOS ONE

**Additional Editor Comments:**

I invited and received three reviews for your manuscript. All reviews raised some substantial concerns about your manuscript as it currently stands. As such, I would recommend “major revision”. I would kindly ask you to go through all comments raised by each reviewer and address them properly before sending a revised version of this manuscript. Please check all PLOS ONE style requirements available via https://journals.plos.org/plosone/s/submission-guidelines before submitting the revised version. 

2. We note that Figures 1, 3 and 5 in your submission contain map/satellite images which may be copyrighted.

a. You may seek permission from the original copyright holder of Figures 1, 3 and 5 to publish the content specifically under the CC BY 4.0 license.

**Reviewers' comments:**

Reviewer's Responses to Questions

**Comments to the Author**

1. Is the manuscript technically sound, and do the data support the conclusions?

Reviewer #1: Yes

Reviewer #2: Yes

Reviewer #3: Yes

2. Has the statistical analysis been performed appropriately and rigorously? 

Reviewer #1: Yes

Reviewer #2: Yes

Reviewer #3: No

3. Have the authors made all data underlying the findings in their manuscript fully available?

Reviewer #1: Yes

Reviewer #2: Yes

Reviewer #3: No

4. Is the manuscript presented in an intelligible fashion and written in standard English?

Reviewer #1: Yes

Reviewer #2: Yes

Reviewer #3: No

5. Review Comments to the Author

Reviewer #1: The submitted Ms deals with an important topic and presented in very good shape that reflects the authors’ effort in planning and carrying it. However, very few comments and corrections were inserted within the text, these are:

1. Page 5, Line 92: are least susceptible Delete

2. Page 5, Line 96: Change “be a reported” to reported

3. In the whole text, replace/write Jazan for state, Jizan for city, for examples:

a. Change “Jizan” to Jazan in “Page 9, Line 186; Page 10, Line 193; Page 10, Line 193; Page 10, Line 195”

b. Page 13, Line 264: Jazan to Jizan

4. Page 10, Line 192: Change “reported to” reported that

5. Page 11, Line 223: (Fig 4) delete, repetition

6. Page 13, Line 267: Change “suggested to” suggested that

7. Page 14, Line 281: Change “arabiens” to arabiensis

8. Page 14, Line 281: and not Italic

9. Since mosquito control is the responsibility and taken by Health authorities:

a. Change Ministry of Agriculture to Ministries of Agriculture and Health (Page 14, Line 288)

b. Change “Agriculture” to Health (Page 16, Line 328)

Reviewer #2: The authors evaluate the 2000 Rift Valley fever virus outbreak in Yemen and Saudi Arabia using satellite data to determine favourable conditions based on rainfall and water availability for mosquitos. The authors further evaluate conditions from 2000 to 2018 to identify favourable conditions for Rift Valley fever virus outbreaks in the region.

Minor comments

Line 94 Replace with There is variability in the susceptibility of different species and breeds with sheep, goats, cattle and camels being the primary species affected.

Can this satellite data be used to determine the risk of Rift Valley fever virus outbreaks in regions where the disease is cyclically present such as South Africa and Kenya?

This is important as if this is the case this could be used to potentially predict Rift Valley fever virus outbreaks and be used to allow vaccination of animals prior to outbreaks.

Reviewer #3: The authors anticipated RVF outbreak in Saudi Arabia using remote sensing data. The manuscript is well-written; however, I have some concerns to the manuscript as it currently stands. These points should be addressed before considering a revised version of this manuscript. 

- Provide justification for using this very finer spatial resolution for your study. Do you have RVF data in such spatial resolution for comparison. 

- Similar analyses was presented early in 2014 ASTMH meeting by Egyptian researcher working at the University of Kansas (Check http://tiny.cc/95r2cz). I remember that Fig 1 was presented by Samy during the meeting, particularly the close-up map to Saudi Arabia with prediction to the West coast of Saudi Arabia and Yemen. Authors should refer clearly to this study across the manuscript text and cite it to the reference section.   

- How authors managed to scale different spatial resolution of diverse climate data to present data with a finer resolution. 

- These models didn't consider any additional dimensions for the key roles of the vector in anticipating the location with suitable conditions for disease occurrence. 

- Robust validation approach for your prediction; i think authors should provide sensitive tools to evaluate their results. Positive cases identified in Western Saudi Arabia were more than just 4 sites presented in Fig 1 and 3. 

6. PLOS authors have the option to publish the peer review history of their article (what does this mean?). If published, this will include your full peer review and any attached files.

Reviewer #1: Yes: Mohamed A Kenawy

Reviewer #2: No

Reviewer #3: No

---

## [Author Response · Author response to Decision Letter 0]

30 Apr 2020

We agree with all the points and suggestions of the reviewers and have addressed every one of their concerns or questions:

1. We have studied PLOS ONE's style requirements and have modified our manuscript accordingly;

2. Figures 1 and 3 are figures which Jennifer Small, one of our coauthors, produced. Because she is employed by NASA, these images are in the public domain and are not copyrighted. Katherine Melocik, another coauthor produced figure 5. Figure 5a is copyrighted and we have obtained permission from the National Geospatial Intelligence Agency for public release of figure 5a--it is labeled accordingly. Because figure 5b is a derived product from NASA-licensed commercial satellite data, this figure is not copyrighted. 

3. All data used in our manuscript are publicly available. This includes the Landsat-7 data in figure 3, the AVHRR data in figure 4a, the MODIS data in figure 4b, figure 5a, and the digital DEM in figure 5b.

4. We have amended our list of authors ensuring that each author is linked to an affiliation.

Reviewers' comments:

Reviewer #1: We have heeded this reviewers comments and suggestions for every one of his nine numbered points. We thank this reviewer for his thoroughness.

Reviewer #2: We replaced line 94 with exactly the wording suggested by this reviewer. We also have made excellent progress using satellite data to predict Rift Valley fever virus outbreaks in areas such as Kenya and South Africa to allow vaccination of animals prior to outbreaks.

Reviewer #3: We use 2 m WorldView-1 satellite data in figure 5 because these data are a stereo pair. These data enable the 2 m scale 3-D mapping of inundation potential for mosquito breeding habitats. We do not have Rift Valley fever data at the same spatial scale. However, our point here is to show a new potential for mapping fine-scale mosquito breeding habitats.

We thank reviewer #3 for bringing to our attention the work of Samy, Kenawy, and Townsend (2014). Our NASA librarian was able to find a description of this poster from the 2014 ASTMH meeting. We concur with reviewer #3 that this reference is very similar to our work and cite this work accordingly as reference 39 in our revised manuscript. We give this reference credit for their earlier description of the methodology for what we have also done. Furthermore, we acknowledge that our work strengthens their earlier work by using additional satellite data available after 2014.

We were able to scale different spatial resolutions of diverse climate data by using 30 m Landsat data as in figure 3, compared to the 250 m MODIS data and 8 km AVHRR data we used in figure 4.

We acknowledge Samy, Kenawy, and Townsend (2014) are the masters of ecological niche modeling which we did not use. Instead, we focused on uses of satellite data that support their expertise in ecological niche modeling. We showed very similar NDVI results using Landsat, MODIS, and AVHRR data sets over a 38-year period at three different spatial scales. Lastly, we show the use of stereo satellite data to map inundation potential at the 2 m scale. This technique compliments ecological niche modeling where inundation is an important consideration for ecological niches.

We attempted to use robust validation tools to test our Arabian predictions. See lines 292 to 315 in our revised manuscript. However, the results of these validations were mixed and inconclusive. This led us to conclude that Saudia Arabia had instituted a thorough and continuing program of medical and veterinary Rift Valley fever surveillance since the 2000 outbreak.

---

## [Editor Report · Decision Letter 1]

4 May 2020

Reanalysis of the 2000 Rift Valley fever outbreak in Southwestern Arabia

PONE-D-19-21159R1

Dear Dr. Tucker,

We are pleased to inform you that your manuscript, "Reanalysis of the 2000 Rift Valley fever outbreak in Southwestern Arabia" (PONE-D-19-21159R1), has been judged scientifically suitable for publication and will be formally accepted for publication once it complies with all outstanding technical requirements.

With kind regards,

Abdallah M. Samy, PhD

Academic Editor

PLOS ONE

---

## [Editor Report · Acceptance letter]

3 Dec 2020

PONE-D-19-21159R1 

Reanalysis of the 2000 Rift Valley fever outbreak in Southwestern Arabia 

Dear Dr. Tucker:

I'm pleased to inform you that your manuscript has been deemed suitable for publication in PLOS ONE. Congratulations! Your manuscript is now with our production department. 

Kind regards, 

on behalf of

Dr. Abdallah M. Samy 

Academic Editor

PLOS ONE